# Activation of the NALP3-CASP1-IL-1 β Inflammatory Pathway by Pesticide Exposure in Human Umbilical Vein Endothelial Cells

**DOI:** 10.3390/ijms26104947

**Published:** 2025-05-21

**Authors:** Antonella Mazzone, Ylenia Della Rocca, Federica Flamminii, Simone Guarnieri, Dainelys Guadarrama Bello, Antonio Nanci, Oriana Trubiani, Francesca Diomede, Jacopo Pizzicannella

**Affiliations:** 1Department of Innovative Technologies in Medicine and Dentistry, University “G. d’Annunzio” Chieti-Pescara, 66100 Chieti, Italy; antonella.mazzone@unich.it (A.M.); ylenia.dellarocca@unich.it (Y.D.R.); oriana.trubiani@unich.it (O.T.); 2Laboratory for the Study of Calcified Tissues and Biomaterials, Department of Stomatology, Faculty of Dental Medicine, Université de Montréal, Montréal, QC H3C3J7, Canada; dainelys.guadarrama.bello@umontreal.ca (D.G.B.); antonio.nanci@umontreal.ca (A.N.); 3Department for the Promotion of Human Sciences and Quality of Life, San Raffaele Roma Open University, Via di Val Cannuta 247, 00166 Rome, Italy; federica.flamminii@uniroma5.it; 4Department of Neuroscience, Imaging and Clinical Sciences, “G. d’Annunzio” University of Chieti-Pescara, Via Luigi Polacchi 11, 66100 Chieti, Italy; simone.guarnieri@unich.it (S.G.); jacopo.pizzicannella@unich.it (J.P.); 5Department of Biochemistry and Molecular Medicine, Faculty of Medicine, Université de Montréal, Montréal, QC H3C3J7, Canada

**Keywords:** HUVECs, pesticides, inflammatory pathway, immunofluorescence, RT-PCR, SEM, ROS

## Abstract

Barrier function regulation, angiogenic potential, and immune response modulation are only a few of the many roles of the vascular system that nowadays represent one of the main targets for environmental pollutants, in particular, pesticides. We have used human umbilical vein endothelial cells (HUVECs) as an in vitro model to investigate the effects of pesticides on the activation of the NALP3-CASP1-IL-1β inflammatory pathway using real time PCR (RT-PCR) and immunofluorescence investigations, reactive oxygen species (ROS) generation, and morphological alterations with scanning electron microscopy (SEM) analysis. Our findings offer a comprehensive evaluation of the cellular and molecular damage induced by pesticide exposure and show strong inflammasome activation. They indicate that these chemicals may initiate necroptosis and drive prolonged inflammation in endothelial cells. This study provides crucial insights into how pesticides contribute to endothelial dysfunction, highlighting the need for further investigation into their inflammatory and immune-modulatory effects on vascular health.

## 1. Introduction

Pesticides have played a crucial role in advancing human activities, particularly in agricultural production, preservation, pest control, and disease management; the extensive adoption of pesticides throughout the past century can be attributed to the proliferation of new, powerful organic chemical agents. However, their inherent toxicity to living organisms, while advantageous in agricultural contexts, inevitably raises concerns about their impact on humans and other animal species. Extensive literature links pesticide exposure to various human health issues, including neurotoxicity, mutagenicity, carcinogenicity, teratogenicity, and endocrine disruption [1].

In addition, in the last few years there has been a heightened demand for food that resulted in an increased human exposure to pesticides; this can occur through various pathways, including occupations associated with pesticide production, transportation, application, and distribution, as well as residence in areas marked by heightened pesticide residue, and the proliferation and accumulation of pesticides within the food chain. This issue is of significant concern considering the mounting epidemiological and experimental evidence establishing an association between pesticide exposure and the prevalence of diverse health ailments in human populations.

Many believe that pesticides can be harmful to humans not only in large amounts causing acute poisoning, but also in small amounts or when combined; the effects of multiple pesticides together can be stronger or weaker than expected based on dosage [2].

Despite continuous long-term exposure to low levels of pesticides, the potential combined risks have not been thoroughly investigated, with only a few studies addressing this issue.

Due to the lack of scientific evidence on the safety of prolonged exposure to combinations of pesticide residues, it is crucial to thoroughly investigate and accurately predict the risks associated with multiple exposures, beginning with in vitro evaluations of pesticide mixtures from different chemical families. The European Food Safety Authority (EFSA) has highlighted that nearly 27% of fruits and vegetables in Europe are simultaneously contaminated by multiple pesticides.

One of the main targets of pesticides is the vascular system, representing the first line of contact with circulating substances, including environmental contaminants as pesticides.

Indeed, the sublingual mucosa is highly vascularized [3], and when pesticide-contaminated foods are ingested, endothelial cells in the vasculature are among the first cellular structures to encounter these xenobiotics. This is particularly relevant because the sublingual route allows for rapid absorption of substances into the bloodstream, thereby exposing endothelial cells to potential toxicants at an early stage [4].

Moreover, the vascular system assumes a pivotal role in maintaining blood vessel homeostasis, regulating vessel permeability, and modulating the response of blood vessels to various physiological and pathophysiological stimuli. Therefore, structural and functional aberrations in endothelial cells can significantly contribute to the pathogenesis of vascular wall disorders, including thrombosis, atherosclerosis, and vasculitis [5].

Insecticides and fungicides are the most commonly used categories of pesticides today. There are several groups of pesticides classified based on their chemical composition, including organochlorines, organophosphorus compounds, carbamates, pyrethrins, and pyrethroids [1].

Boscalid (belonging to carboxamides), Pyraclostrobin (classified as strobilurins), Propamocarb (a type of organic nitrogen-carbamates), and Lambda-cyhalothrin (categorized as pyrethroids) are the substances selected after a thorough review of the industrial database, with a focus on the notable remnants found within the past five years due to simulating actual conditions [6].

Boscalid and Pyraclostrobin are fungicides that disrupt the respiratory chain of pathogens, preventing their energy production and cellular functions [7].

Propamocarb is a carbamate fungicide that impacts the production of cell membrane components through biochemical synthesis [8].

Lambda-cyhalothrin is a pyrethroid pesticide that has a broad spectrum of effects, such as inducing oxidative stress in unintended species [9].

This study aims to examine how these pesticides impact human umbilical vein endothelial cells (HUVECs), chosen as an in vitro model to represent the vascular system due to their well-established role in studying vascular biology, inflammation, and toxicity [10].

HUVECs have been the predominant endothelial cell type used in in vitro research since 1973 [11].

Indeed, they present many advantages: technical advantages, being accessible, being easy to handle and reproducible, and other advantages such as being useful to functionally characterize EC lines, to study the role of adhesion molecules (such as E-selectin, platelet endothelial cell adhesion molecule [PECAM-1], cadherin-5, and proteases), and to investigate both the synthesis of extracellular proteins and the blood vessel maturation [12].

Therefore, HUVECs, physiologically representing the human vascular endothelium, can be used as an in vitro model to study the physiological and pathological effects of different stimuli both in an isolated form, and in co-culture with other cell types [10].

Moreover, the HUVEC model can better represent some aspects of human disease as compared to animal models [10].

According to those affirmations, HUVECs represents the in vitro model that more likely mirrors human EC behavior as compared to cell lines [13].

This study specifically aims to investigate the NALP3/CASP1/IL-1β inflammatory pathway on the in vitro model HUVEC treated with both individual and combined pesticides.

This pathway regulates many functions and many inflammatory responses of both innate and adaptive immune factors [14,15]. NALP3, caspase 1, and the pro-inflammatory cytokine interleukin-1 beta (*IL-1β*) are the key proteins of the so-called “inflammasome”.

Inflammasomes are composed of three principal components: a scaffold protein, an adaptor protein, and effector proteins [16].

The NALP3inflammosome plays a pivotal role in the activation of pro-caspase-1, facilitating caspase-1 activation, and mediating the processing of the pro-IL-1β in its mature form cytokine that aims to induce the highly inflammatory form of cell death known as pyroptosis [17,18]. Thus, the confirmed activation of this pathway can suggest the consequent activation of this particular cell death [19].

In our study, we thoroughly investigated the activation of the NALP3-CASP1-IL-1β inflammatory pathway and the morphological alterations caused by pesticides on HUVECs by conducting immunofluorescence analysis and RT-PCR, also performed to investigate the expression of other molecules that can be involved in the inflammation. In fact, mRNAs of interleukin 6 (IL-6), of tumor necrosis factor *(*TNF*)* and of CCL2 (MP-1) were evaluated.

Interleukin 6 (IL-6) and tumor necrosis factor (TNF) are key pro-inflammatory cytokines involved in the inflammatory response, playing a significant role in various diseases. IL-6 is a critical mediator in acute inflammation while TNF is considered a primary trigger of the inflammatory cascade [20,21] since it is part of pattern recognition receptors (PRRs) that can induce the transcriptional upregulation of the NALP3 inflammosome [22].

Moreover, the literature suggests a link between the markers IL-1β, IL-6, and TNF; in fact, according to the study “Inflammasome activation at the crux of severe COVID-19”, the activation of IL-1β, released by inflammasome signaling activates monocytes, can lead to the secretion of both IL-6 and TNF [23]. Another study showed an additive interaction between IL-1β and TNFα, depending on CREB binding and H3K14 acetylation that leads to the elevation of IL-6 expression in adipocytes [24].

Moreover, a study published in Frontiers in Immunology examined how IL-6 and CCL2 interact during EMT, highlighting a positive feedback loop mediated by STAT3 that amplifies the inflammatory signaling and promotes cellular invasiveness. In particular, the study observed that IL-6 induces the expression of CCL2, which in turn stimulates EMT, leading to the upregulation of mesenchymal markers [25].

CCL2 is also known as MCP-1 and it acts as a chemoattractant since it attracts leukocytes to sites of inflammation; this molecule helps to recruit leukocytes and potentially disrupt the junctions developed by the cadherins, facilitating leukocyte migration [26,27].

Another important point of the inflammasome is that it can be activated in response to cellular damage that can be derived from multiple molecular and cellular events, including the production of reactive oxygen species (ROS) [18].

In the literature, it is reported that there can be a feedback loop between ROS and NALP3inflammasome [28]; in fact, while ROS represent one of the critical mediators of NALP3inflammasome activation, NALP3 inflammasome itself recruits inflammatory cells, such as macrophages and neutrophils, that can lead to ROS production [28]. Since these studies, ROS generation and SEM analysis have been performed to validate, respectively, the oxidative stress and the morphological alterations derived by pesticides exposure.

## 2. Results

### 2.1. Cell Viability Evaluation (MTS)

The MTS assay demonstrated time-dependent variations in cell metabolic activity between control and pesticide-exposed HUVECs (Figure 1). After 24 h (A) of pesticide-exposure, the treated cells exhibited an increased metabolic activity compared to the control. By 72 h (B), cellular metabolic activity started to be less in treated cells when compared to the control, being more evident after 10 days (C) of treatment in all treated groups, especially in HUVEC cells exposed to the combination b + py + pr and the combination b + py + lc.

### 2.2. Trypan Blue Exclusion Test

Figure 2 shows the cell growth curve obtained from the Trypan blue exclusion test performed on untreated HUVEC (ctrl) and on HUVEC treated with pesticides for 24 h, 72 h, 10 days.

### 2.3. Genes Expression (RT-PCR)

The histograms illustrate *NALP3* (Figure 3), *CASP1* (Figure 4), *IL-1β* (Figure 5), *IL-6* (Figure 6), *TNF* (Figure 7) and *CCL2* (Figure 8) genes expression assessed via RT-PCR in HUVEC cultured alone (ctrl), with Boscalid (b), with Pyraclostrobin (py), with Propamocarb (pr), with Lambda-cyhalothrin (lc), with Boscalid+ Pyraclostrobin (b + py), with Propamocarb+ Lambda-cyhalothrin (pr + lc), with Boscalid+ Pyraclostrobin+ Propamocarb (b + py + pr), and with Boscalid+ Pyraclostrobin+ Lambda-cyhalothrin (b + py + lc) for 10 days.

### 2.4. Immunofluorescence Analysis

The fluorescence images display the presence of NALP3/CASP1/IL-1β (Figure 9, Figure 10 and Figure 11) in HUVEC cultured alone (A1), with Boscalid (A2), Pyraclostrobin (A3), Propamocarb (A4), Lambda-cyhalothrin (A5), Boscalid + Pyraclostrobin (A6), Propamocarb + Lambda-cyhalothrin (A7), Boscalid + Pyraclostrobin + Propamocarb (A8), and Boscalid + Pyraclostrobin + Lambda-cyhalothrin (A9) for 10 days.

The study showed a notable increase in NALP3 (Figure 10), in CASP1 (Figure 11) and in IL-1β (Figure 12) protein expression in HUVECs exposed to pesticides for 10 days compared to those cultured without pesticides.

Further quantitative analysis of the immunolabeling for NALP3 (Figure 12), CASP1 (Figure 13), and IL-1β (Figure 14) expression in HUVEC cells notably revealed significant differences between the control and the treated groups. Compared to the control, all tested pesticides caused a reduction in inflammasome markers’ expression, confirming the qualitative picture of IF.

### 2.5. R0S (Reactive Oxidative Stress)

The ROS probe 2′, 7′-dichlorodihydrofluorescein diacetate (DCFH-DA) was employed to identify the reactive oxygen species (ROS) created in HUVECs cultured by itself and in HUVECs cultured with pesticides. Intracellular esterases broke down the acetate groups of DCFH-DA, enabling the probe to enter cells and distribute evenly within the cytoplasm and organelles. This led to the transformation of DCFH-DA into the fluorescent DCF.

Confocal microscopy was used to capture the fluorescence images in Figure 15. The HUVECs cultured alone (A1) showed a reduced fluorescence signal in contrast to the ones treated with pesticides (A2–A9), suggesting a clear rise in ROS production in the treated HUVECs.

Figure 16 shows the statistical analysis obtained from quantifying the ROS fluorescence by ROI areas measured.

### 2.6. Cell Morphology

SEM analysis at 10 days of treatment revealed significant differences in shapes in the HUVEC control compared to HUVECs treated with pesticides.

Treated HUVECs (Figure 17B–I) present a change in their physiological morphology and a reduction in blebbing when compared to HUVECs control (Figure 17A).

## 3. Discussion

The increase in population in the last few years has led to a higher demand of food and, therefore, to higher human exposure to pesticides; this issue nowadays involves both urban and agriculture environments [29]. The importance of those pollutants in preventing food safety is well known; however, while advantageous in agricultural contexts, inevitably there are concerns about their impact on humans and other animal species [1].

In fact, much evidence has established an association between pesticide exposure and the prevalence of different health issues in human populations, including neurotoxicity, mutagenicity, carcinogenicity, teratogenicity, and endocrine disruption [1].

Our study highlights the modulation of the NALP3/CASP1/IL-1β inflammatory pathway on HUVECs; since 1973, HUVECs have represented an optimal model to best reproduce the vascular endothelium [11].

The endothelium tissue covers many roles, as barrier function regulation, angiogenic potential, and immune response modulation and so on [30].

This study investigates this tissue because is one of the first involved in the exposure with the pesticides due to the high vascularization that is in the sublingual area [3].

Boscalid (carboxamides), Pyraclostrobin (strobilurins), Propamocarb (organic nitrogen-carbamates), and Lambda-cyhalothrin (pyrethroids) are the pesticides chosen for this study, after a deep investigation trough the European database (EFSA). The high natural daily intake (hNDI) is the value used for the study [31]; it indicates the amount ingested from food and it is calculated on the basis of the data presented by Leveque et al. (2019) [32].

Our investigation into the effects of those pesticides on HUVECs, both individually and in combination, provides critical insights into their role in endothelial dysfunction and inflammatory responses.

Our results indicate that pesticide exposure leads to significant alterations in endothelial cell physiology, potentially compromising vascular homeostasis.

When two or more pesticides are used together, their combined effects could be stronger than expected [33]; this highlights the importance of thoroughly addressing the potential risks associated with pesticide exposure.

The MTS assay demonstrated time-dependent variations in cell viability between control and pesticide-exposed HUVECs (Figure 1). After 24 h of pesticide exposure, treated cells exhibited increased metabolic activity compared to the control, likely because the cells were attempting to respond to the treatment stress. This adaptive response could reflect a transient attempt to counteract pesticide-induced toxicity through enhanced energy production and cellular defence mechanisms. However, by 72 h, cell activity began to decline in the treated cells, becoming more evident after 10 days; particularly, the pesticides Boscalid and Lambda-cyhalothrin decreased the metabolic activity of HUVECs when compared to Propamocarb and Pyraclostrobin. Interestingly, the combination of Boscalid + Pyraclostrobin resulted in a more significant reduction in metabolic rate compared to the combination of Propamocarb + Lambda-cyhalothrin. Notably, both the combinations Boscalid + Pyraclostrobin + Propamocarb and Boscalid + Pyraclostrobin + Lambda-cyhalothrin led to a higher rate metabolism rate compared to other experimental points.

The Trypan blue exclusion test (Figure 2) showed that, among the individual pesticides tested, b, py and lc demonstrated a stronger inhibitory effect on HUVEC proliferation compared to pr. These differences may reflect variations in the mode of action or cellular uptake of the compounds. The mixture of pesticides determined a lower cell proliferation compared to the single pesticides; among the combinations, bpypr and bpylc led to the most reduction in cell quantity for all the time points. In general, in all the experimental points, it is observable that treating the cells for 10 days led to the lowest results. This suggests the time-dependent effect of pesticide exposure.

Notably, RT-PCR evaluation confirmed the involvement of the inflammasome NALP3, which suggests a mechanistic link between pesticide exposure and endothelial inflammation. This aligns with previous research indicating that chronic pesticide exposure may contribute to the development of vascular disorders, including atherosclerosis and thrombosis.

In fact, Figure 3, Figure 4 and Figure 5 showed the activation of the mRNA of the markers *NALP3*, *CASP1*, and *IL-1β* demonstrating that both single and combinations of pesticides induced higher expression of mRNA levels compared to the control cells. In particular, among the single pesticides, Boscalid and Lambda-cyhalothrin were the most inflammatory; all the combined treatments showed the most significant results when compared to the other experimental points, in particular, the combination of the three pesticides (b + py + pr, b + py + lc). The same results are observable from the m RNA values of the markers *IL-6* (Figure 6) and *TNF* (Figure 7), whose activation confirmed the pro-inflammatory process switched on by pesticide exposure. In addition, the expression of the marker *CCL2* (Figure 8) suggests the procoagulation phenome induced by the pollutants. This marker confirms the trend observed before; in fact, it still shows its higher level of expression in HUVECs treated with the combination of pesticides compared to the single, in particular, when three pesticides are together, confirming their synergic effect.

Fluorescence microscopy revealed the presence of NALP3, CASP1, and IL-1β proteins in HUVECs cultured alone and with pesticides for 10 days, following the results obtained from the RT-PCR; in fact, the combined pesticide treatment still elicited the greatest increase in their expression (Figure 9, Figure 10 and Figure 11).

Quantitative analysis was obtained through ROI measurements (Figure 10, Figure 11 and Figure 12).

All these findings suggest the additive or synergistic effects induced by the pesticide mixture.

ROS production was assessed using the DCFH-DA probe, which detected an increased fluorescence signal in pesticide-treated HUVECs compared to controls, indicating heightened oxidative stress. The results followed the same trend as the immunofluorescence data, reinforcing the link between pesticide exposure and cellular stress responses (Figure 15 and Figure 16). However, it is noteworthy that the combination of Pyraclostrobin and Lambda-cyhalothrin (A7, Figure 15) appeared to induce a relatively lower oxidative stress response compared to other pesticide combinations.

ROS results suggest the loop phenome described before that is between the inflammasome activation and ROS production.

Furthermore, SEM analysis revealed alterations in pesticide-treated HUVECs, including the loss of physiological morphology and the presence of blebbing, further supporting the notion of pesticide-induced cellular stress and dysfunction (Figure 17)

Moreover, changes in the cellular blebbing have been observed between control and treated cells that showed a loss in that cellular phenomenon, in particular, observed in the combinations pr + lc (G, Figure 17), b + py + pr (H, Figure 17), b + py + lc (I, Figure 17).

This result could lead to important alterations in the communication between the endothelium and the other tissues. In fact, being the “endothelium”, the single cell-lining tissue formed by the endothelial cells in the blood and lymphatic vessels, it controls the exchange of oxygen and nutrients between the vessel contents and underlying tissues [34]. Therefore, having the “front lines” role of vessels, endothelial cells are exposed to circulating cells and plasma releasing a significant proportion of the extracellular vesicles (EVs) [35].

EVs are a widespread form of intercellular communication occurring under physiological and disease states [36,37]; EVs released by the endothelium are at the base for supporting vascular homeostasis, including maintenance of the antithrombogenic surface of the vessels (blood fluidity) and vasodilation, inhibition of inflammation, cell survival and angiogenesis [38]. Therefore, a loss in the production of the EVs could lead to problems that involve many tissues. Future study will deeply evaluate the RNA profiling of the vesicles to further understand how pesticides can influence the communication between the cells when they are exposed to the pollutant rather than when they are not.

Another aim for future researchers should explore the long-term consequences of pesticide exposure on endothelial function in vivo, as well as the potential reversibility of the observed effects. Additionally, elucidating the specific molecular pathways underlying pesticide-induced toxicity could pave the way for the development of targeted therapeutic interventions to mitigate vascular damage associated with pesticide exposure.

## 4. Materials and Methods

### 4.1. Cell Culture Establishment

HUVECs (C-003-5C), acquired from Invitrogen Life Technologies (Carlsbad, CA, USA), were thawed following the guidelines provided by the manufacturer, and made to grow with Human Large Vessel Endothelial Cell Basal Medium (formerly Medium 200) (M200500) (Invitrogen, Carlsbad, CA, USA) enriched with Low Serum Growth Supplement (S00310) (Invitrogen, Carlsbad, CA, USA) and 10% fetal bovine serum (FBS; Invitrogen, Carlsbad, CA, USA).

When the cells were in their third passage, they were used for the experiment.

### 4.2. Experimental Study Design

The pesticides Boscalid (AST1Y6P006) and Pyraclostrobin (AST1Y6D573) were procured by Lab Instruments S.R.L (Castellana Grotte, BA, Italy), while Propamocarb (45638) and Lambda-cyhalothrin (31058) were obtained from Sigma Aldrich (St. Louis, MO, USA).

The experimental points were performed in triplicate:HUVECs cultured alone (ctrl);HUVECs cultured with Boscalid (b);HUVECs cultured with Pyraclostrobin (py);HUVECs cultured with Propamocarb (pr);HUVECs cultured with Lamba-cyhalothrin (lc);HUVECs cultured with the combination Boscalid+ Pyraclostrobin (b +py);HUVECs cultured with the combination Propamocarb+ Lambda-cyhalothrin (pr +lc);HUVECs cultured with the combination Boscalid+ Pyraclostrobin+ Propamocarb (b + py + pr);HUVEC cultured with the combination Boscalid+ Pyraclostrobin+ Lambda-cyhalothrin (b+ py + lc).

The cells were grown for 10 days, except for the MTS and Trypan blue exclusion test, where the treatment was applied for three distinct periods: 24 h, 72 h, and 10 days.

### 4.3. Calculation of Application Doses and Preparation of Pesticide Solutions

For pesticide concentration, the high nutritional daily intake (hNDI) values were determined on the methods outlined in the studies by Hochane et al. (2017) [39] and Leveque et al. (2019) [32]; this value represents the actual exposure of the French population to each pesticide [32].

It is computed from the Maximum Residue Levels (MRLs) in food commodities established by the EFSA [40]; to determine chronic exposure, the EFSA calculates the hNDI value following a methodology similar to that of the Theoretical Maximum Daily Intake (TMDI):TMDI = ∑(MRLi × Fi)
where MRLi denotes the maximum residue level in a particular food, and Fi represents food consumption rates. Given that vegetables have a higher dietary prevalence than fruits, the MRL data were examined in the EFSA database that provides information about the MRL values of pesticides. Consumption rates (Fi) were obtained from national (CREA Scientific Council, 2018) and international (WHO) dietary recommendations, which suggest a minimum daily intake of 400 g of fruits and vegetables. Considering that people usually consume portions of fruits and vegetables twice a day, EFSA data report that the general total intake is of 860 g daily, 70% of which should be vegetables.

The value of hNDI, extrapolated from all diets [41], was converted first to mg/kg. bw/day and then to g/L and then in mol/L for in vitro studies.

The total absorption of the residue concentrations and their dilution in 5 L of blood in a 60 kg individual were taken into account to determine the blood concentration (mg/L and μmol/L) to which different organs might be theoretically exposed.

Pesticides powders were dissolved in 1 mL of Dimethyl sulfoxide (DMSO) solution; using the HUVECs’ medium, pesticides were then concentrated at their hNDI values. The maximum of the DMSO final concentration was 0.2%.

This concertation of DMSO was used for the control condition represented by HUVECs treated without pesticides.

The choice of mixture type was based on a thorough examination of different pesticide combinations found in agricultural commercial formulations. The cumulated impact is calculated by adding together the values listed in Table 1.

Since it is widely acknowledged that pesticides can have detrimental effects on human health, not only at high doses resulting in acute poisoning but also at low doses and when combined with other pesticides, individual pesticides and the following combination were chosen for the study: Boscalid + Pyraclostrobin, Boscalid + Pyraclostrobin +Propamocarb, Propamocarb + Lambda-cyhalothrin, Boscalid+ Pyraclostrobin+ Lambda-cyhalothrin, Boscalid+ Pyraclostrobin+ Propamocarb + Lambda-cyhalothrin (see Table 2). 

### 4.4. Metabolic Activity Evaluation(MTS)

The 3-(4,5-dimethylthiazol-2-yl)-5-(3-carboxymethoxyphenyl)-2-(4-sulfo-phenyl)-2H-tetrazolium (MTS) assay was used to measure the metabolic activity of HUVECs cultured under different conditions, including exposure to pesticides, in a study conducted with the CellTiter 96^®^ Aqueous One Solution Cell Proliferation Assay from Promega, Madison, WI, USA. 3,000 cells were placed in each well of 96-well plates and were grown for 24, 72 h, and 10 days at 37 °C. After each time point, 20 µL of MTS dye solution was added to each well and incubated for 3 h at 37 °C. The Synergy™ HT Multi-detection microplate reader from Biotech in Winooski, VT, USA, was used to measure the formazan product, correlating directly with the number of viable cells in culture, by taking absorbance readings at 490 nm. The MTS assay was replicated in three separate trials.

### 4.5. Trypan Blue Exclusion Test

Trypan blue exclusion test was performed to evaluate the cell proliferation incubating all samples with trypan blue solution at the same endpoints used for the MTS test (24 h, 72 h, and 10 days); subsequently, the cells were counted by light microscopy (Leica DM Il LED fluo, Wetzlar, Germany), using a Burker’s chamber as previously described [42].

### 4.6. RNA Isolation and Real-Time RT-PCR Analysis

The PureLink RNA Mini Kit, obtained from Ambion, Thermo Fisher Scientific in Milan, Italy, was utilized to isolate total RNA, following the manufacturer’s instructions. This RNA was later used to analyze the expression of *NALP3*, *CASP1*, *IL-1β*, *IL-6*, *TNF* and *CCL2* mRNAs using real-time PCR.

To perform reverse transcription into cDNA, one microgram of total RNA was required when utilizing M-MLV reverse transcriptase (Sigma-Aldrich, St. Louis, Missouri, USA), following the manufacturer’s instructions which included sequential incubation at 70 °C for 10 min, 37 °C for 50 min, and 90 °C for 10 min.

The Mastercycler realplex system from Eppendorf, based in Hamburg, Germany, was used to perform real-time PCR with three separate biological replicates for each sample.

Comparison was made between mRNA expression levels of *NALP3* (Hs.PT.58.39303321, Tema Ricerca Srl, Castenaso, BO, Italy), *CASP1* (Hs.PT.56a.22811633.g), *IL-1β* (Hs. PT.58.1518186), *IL-6* (Hs. PT. 58. 40226675), *TNF* (Hs. PT. 58. 45380900), and *CCL2* (Hs. PT.58. 45467977) (see Table 3) in HUVECs culture conditions, with and without pesticides, also using PrimeTime™ Gene Expression Master Mix (cat. n°1055772, Tema Ricerca Srl) according to standard protocols. 

The GAPDH (Hs. PT.39a.22214836) obtained from Tema Ricerca Srl was utilized for normalization as the endogenous control.

The amplification process started with pre-incubation at 95 °C for 3 min, then continued with 40 cycles of denaturation at 95 °C for 15 s and annealing at 60 °C for 1 min. A melting curve analysis was conducted for each run between the temperatures of 60 °C and 95 °C. The gene expression levels were determined using the “2^−ΔΔCT^ method”.

Three separate trials, each containing two measurements for each sample, were conducted for the RT-PCR analysis.

### 4.7. Immunofluorescence Analysis

To visualize the protein expression, 10,000 HUVEC cells were grown on sterilized glass coverslips (1943-10012A, Bellco Glass, Vineland, NJ, USA); after the treatment with pesticides of 10 days, the cells were fixed with PLP (periodate–lysine–paraformaldehyde) solution in PB for 30 min at 4 °C. After this time, cells were washed 3 times with PB for 5 min and permeabilized with 0.5% Triton X-100 (9474680K, LKB-Produkter AB, Bromma, Sweden) in PB for 10 min; 5% skim milk in PB was added to block nonspecific for 1 h.

Primary monoclonal antibodies against human NALP3 (dilution: 1:400) (sc-8422, Santa Cruz Biotechnology, Santa Cruz, CA, USA), CASP1 (dilution: 1:400) (sc-17810, Santa Cruz Biotechnology), and IL-1 β (1:400) (sc-32294, Santa Cruz Biotechnology) were diluted in 0.5% non-fat milk solution and the samples were incubated for 2 h in a humidified environment at RT in the dark.

After 3 washes for 5 min each in PB, Alexa Fluor 488-conjugated goat anti-rabbit secondary antibody (1:400, 1851447, Life Technologies) was diluted in 0.5% of blocking solution, performing the incubation in the same conditions as before; simultaneously, rhodamine-phalloidin (1:150, 2892779, Life Technologies) was added for the duration of this incubation.

Cell nuclei were stained and the coverslips were mounted with mounting medium containing DAPI (Prolong antifade 40,6-diamidino-2-phenyl-indole, dihydrochloride, Molecular Probes, Invitrogen) on glass slides.

The samples were analyzed with a Zeiss Axio Imager M2 Optical Microscope (Carl Zeiss, Jena, Germany), with an oil immersion 63X objective to study the cytoskeleton distribution and the inflammatory markers.

In addition to untreated cells as controls, the differential immunolabeling obtained with the multiple antibodies used provides positive controls for the secondary antibody.

The immunofluorescence assay was repeated in three separate experiments.

### 4.8. Reactive Oxygen Species (ROS) Evaluation

Endothelial cells (HUVEC) were placed in a 35 mm imaging dish (µ-Dish, ibidi GmbH, Gräfelfing, Germany) at a density of 85,000 cells per well and exposed to pesticides for 10 days. Afterwards, the medium was changed to Normal External Solution (NES) with a composition of 125 mM NaCl, 5 mM KCl, 1 mM MgSO_4_, 1 mM KH_2_PO_4_, 5.5 mM glucose, 1 mM CaCl_2_, and 20 mM HEPES at a pH of 7.4.

Next, the cells were exposed to a culture medium containing 10 μM of 2′,7′-dichlorodihydrofluorescein diacetate (H2DCFDA, Thermo Fisher, Waltham, MA, USA) for 30 min at 37 °C, then rinsed with NES. For visualization, only NES was introduced into the plate.

Confocal microscopy (Zeiss LSM800 microscope, Carl Zeiss, Jena, Germany) was used to capture images with consistent acquisition settings using a motorized table named SMC 2009 and the multiple single position acquisition function (Tiles-Advanced setup, carrier 35 mm petri dish) of Zen Blue software (Zen 3.0 SR, Carl Zeiss); the inverted Axio-observer D1 microscope and a W-Plan-Apo 40× 1.3 DIC objective (Carl Zeiss, Jena, Germany) has been used for ROS detection using an excitation of 488 nm. 

Three separate trials were conducted for ROS analysis.

Image analysis was performed offline with Fiji distribution of ImageJ (version 1.53c, National Institutes of Health, Bethesda, MD, USA) to measure the average fluorescence intensity (in arbitrary units, F) and the area of the HUVECs cell (in µm^2^) for each cell. The ROS levels are quantitatively represented as the F/µm^2^ ratio.

### 4.9. Cell Morphology (Scanning Electron Microscopy)

Cells were fixed for 1 h at room temperature (RT) in 2,5% glutaraldehyde (EMS, Hatfield, PA, USA) prepared in 0.1 M sodium phosphate buffer (PB; pH 7.2). Samples were postfixed with 1% aqueous osmium solution (1 h at 4 °C) and washed three times (5 min) with PB. Then, cells were dehydrated in increasing aqueous ethanol concentrations (30–100%) and finally dried in a critical point dryer (CPD 300; Leica Microsystems Inc., Ontario, Canada). Cells cultured on glass were coated with a thin layer of carbon before observation to reduce surface charging of the substrate. Imaging was carried out using an ultrahigh-resolution scanning electron microscope (SEM) Regulus 8220 (Hitachi, Ltd., Tokyo, Japan) operated at 1 kV.

### 4.10. Quantification of Fluorescence of Protein Expression and ROS

Protein expression quantification observed in immunofluorescence was performed using ImageJ software, version 8 (National Institutes of Health, Bethesda, MD, USA). Fluorescence intensity was measured by analyzing the captured images, selecting regions of interest (ROIs), and calculating the mean fluorescence intensity. To ensure accurate quantification, background fluorescence was subtracted. Data were then normalized to control samples to evaluate relative protein expression levels.

### 4.11. Statistical Analysis

GraphPad Prism (version X, GraphPad Software, San Diego, CA, USA) is the software used for the statistical analyses executed. Performing three independent experiments, the derived data were expressed as mean ± standard deviation (SD). The ANOVA test followed by Tukey’s multiple comparison test were performed to determine the statistical significance.

A *p*-value of <0.05 was considered statistically significant [43]. Graphs were generated to represent the results, with error bars indicating the variability within replicates.

## 5. Conclusions

Despite the in vitro limitations, our study confirms that pesticides contribute to the development of inflammation; in particular, the combinations of the pesticides developed more inflammation compared to the single pesticides.

Among the many experimental points, the combinations of B + PY + PR as B + PY + LC showed the most highlighted results.

In conclusion, the observed inflammatory responses and toxicokinetic interactions highlight the potential risks posed by pesticide exposure, particularly in the context of mixture effects. Given the widespread presence of pesticide residues in food and water sources, our findings reinforce the importance of continuous monitoring and evaluation of pesticide safety standards to protect human health.

## Figures and Tables

**Figure 1 ijms-26-04947-f001:**
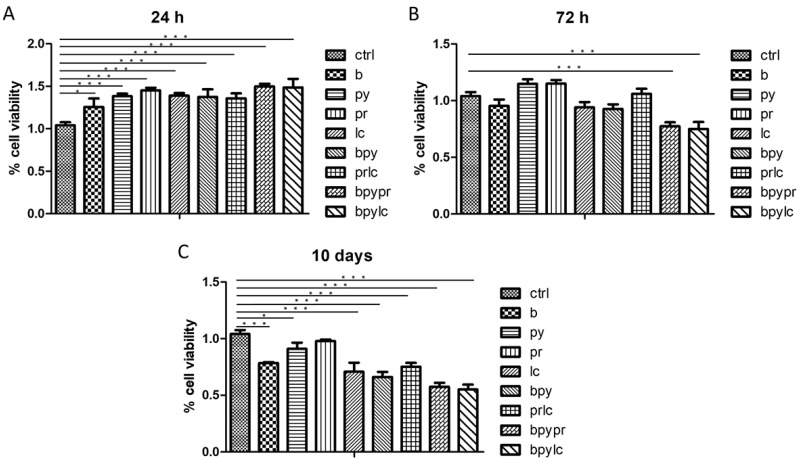
Statistical analysis from the MTS assay show cellular metabolic activity of HUVECs without treatment (ctrl) compared with HUVECs exposed at Boscalid (b), Pyraclostrobin (py), Propamocarb (pr), Lamba-cyhalothrin (lc), Boscalid + Pyraclostrobin (b + py), Propamocarb + Lamba-cyhalothrin (pr + lc), Boscalid + Pyraclostrobin + Propamocarb (b + py + pr), and Boscalid + Pyraclostrobin + Lamba-cyhalothrin (b + py + lc) at different time points: 24 h (**A**), 72 h (**B**), and 10 days (**C**). Data are expressed as mean ± standard deviation (n = 3). Significant differences are indicated (* *p* < 0.05, *** *p* < 0.001). The experiment was conducted three times, each independently.

**Figure 2 ijms-26-04947-f002:**
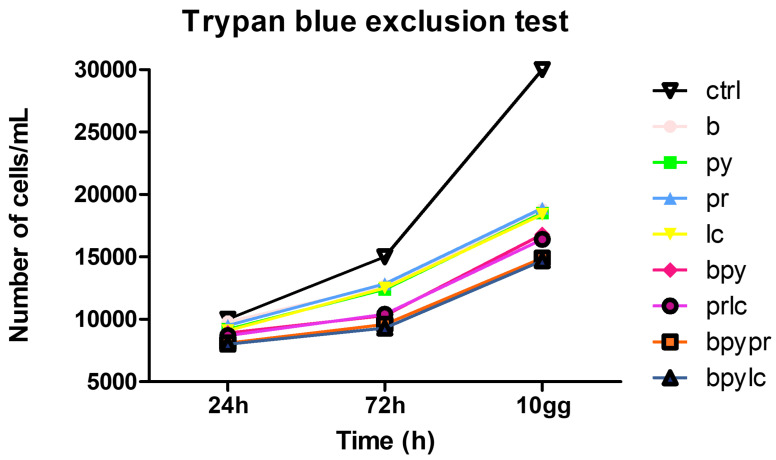
The results of the Trypan Blue exclusion test show the cell growth curve of untreated HUVECs (ctrl) and HUVECs treated with Boscalid (b), Pyraclostrobin (py), Propamocarb (pr), Lamba-cyhalothrin (lc), Boscalid + Pyraclostrobin (b + py), Propamocarb + Lamba-cyhalothrin (pr + lc), Boscalid + Pyraclostrobin + Propamocarb (b + py + pr), and Boscalid + Pyraclostrobin + Lamba-cyhalothrin (b + py + lc) at 24 h, 72 h and 10 days.

**Figure 3 ijms-26-04947-f003:**
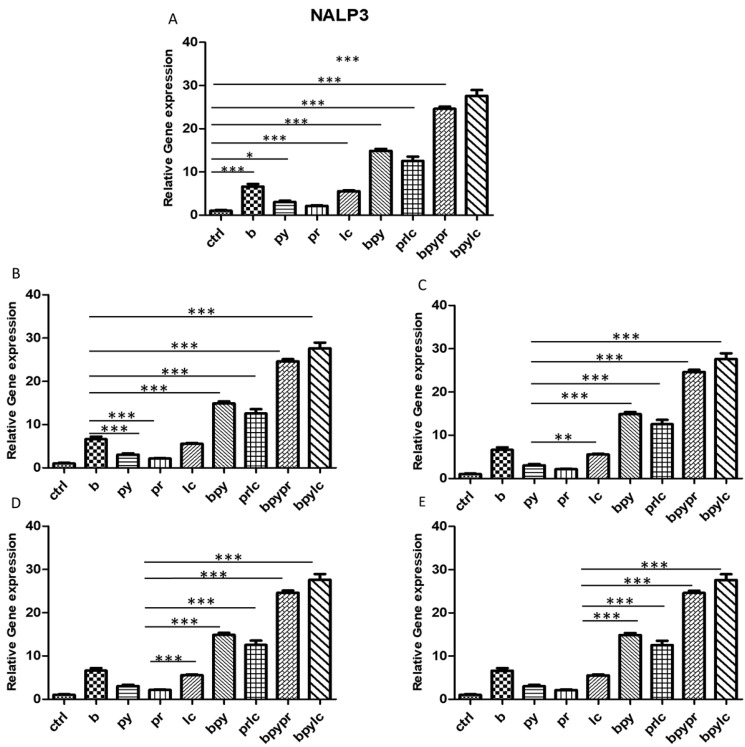
The bar graph shows the statistical analysis, where data are expressed as mean ± standard deviation (SD) from three independent experiments evidencing mRNA levels of *NALP3* between ctrl cells (**A**) and each single pesticide b (**B**), py (**C**), pr (**D**), lc (**E**) with the other experimental points. Statistical significance was determined using one-way ANOVA followed by Tukey’s multiple comparison test. Significant differences compared to the control are indicated (* *p* < 0.05, ** *p* < 0.01, *** *p* < 0.001).

**Figure 4 ijms-26-04947-f004:**
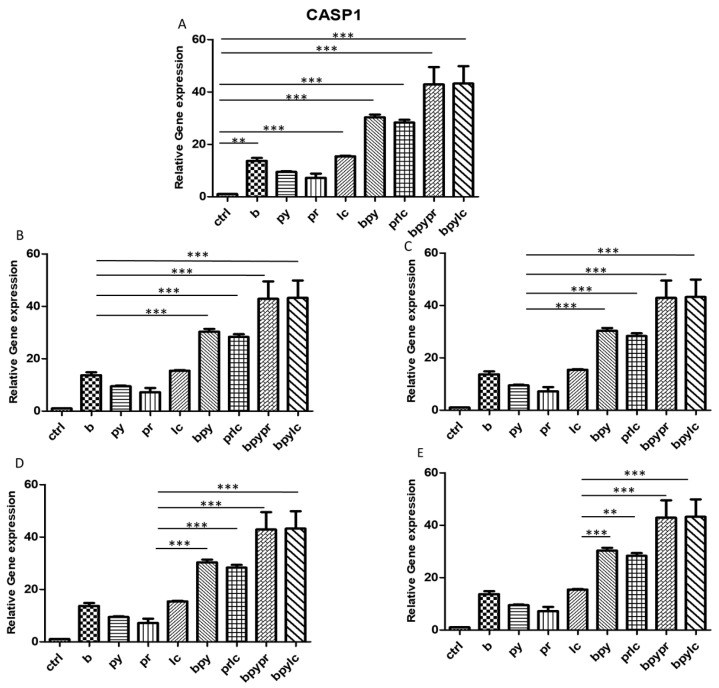
The bar graph shows the statistical analysis, where data are expressed as mean ± standard deviation (SD) from three independent experiments evidencing mRNA levels of *CASP1* between ctrl cells (**A**) and each single pesticide b (**B**), py (**C**), pr (**D**), lc (**E**) with the other experimental points. Statistical significance was determined using one-way ANOVA followed by Tukey’s multiple comparison test. Significant differences compared to the control are indicated (** *p* < 0.01, *** *p* < 0.001). The experiment was conducted in triplicate.

**Figure 5 ijms-26-04947-f005:**
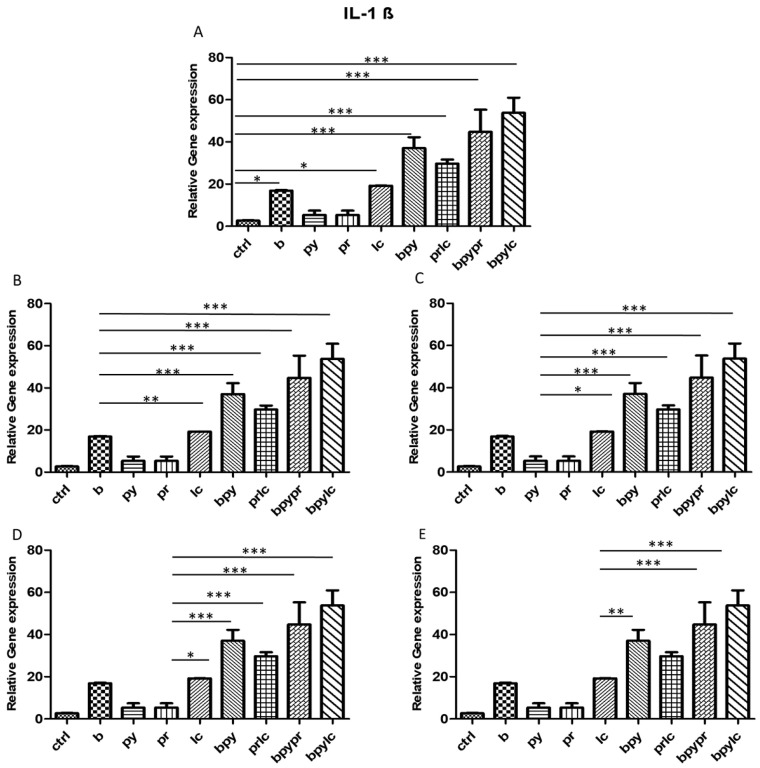
The bar graph shows the statistical analysis, where data are expressed as mean ± standard deviation (SD) from three independent experiments evidencing mRNA levels of *IL-1β* between ctrl cells (**A**) and each single pesticide b (**B**), py (**C**), pr (**D**), lc (**E**) with the other experimental points. Statistical significance was determined using one-way ANOVA followed by Tukey’s multiple comparison test. Significant differences compared to the control are indicated (* *p* < 0.05, ** *p* < 0.01, *** *p* < 0.001).

**Figure 6 ijms-26-04947-f006:**
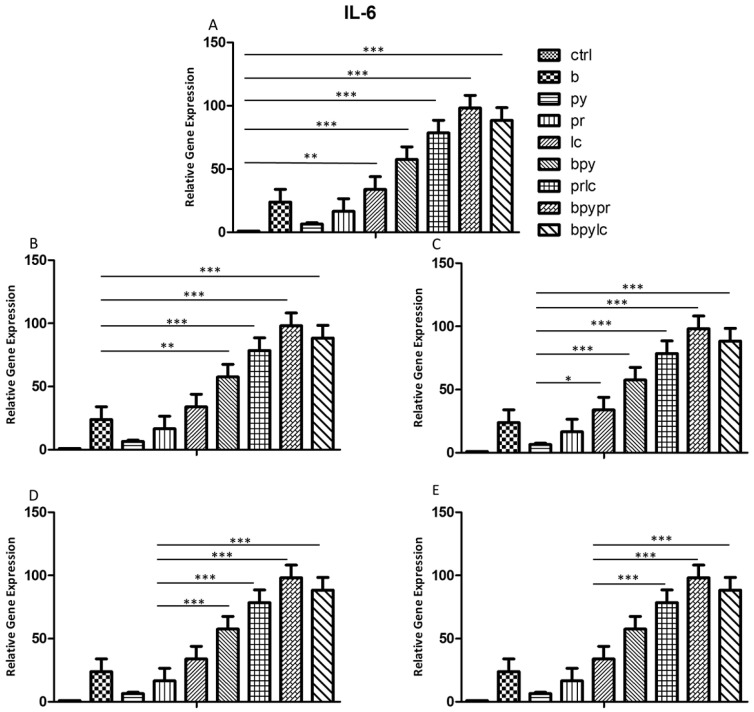
The bar graph shows the statistical analysis, where data are expressed as mean ± standard deviation (SD) from three independent experiments evidencing mRNA levels of *IL-6* between ctrl cells (**A**) and each single pesticide b (**B**), py (**C**), pr (**D**), lc (**E**) with the other experimental points. Statistical significance was determined using one-way ANOVA followed by Tukey’s multiple comparison test. Significant differences compared to the control are indicated (* *p* < 0.05, ** *p* < 0.01, *** *p* < 0.001).

**Figure 7 ijms-26-04947-f007:**
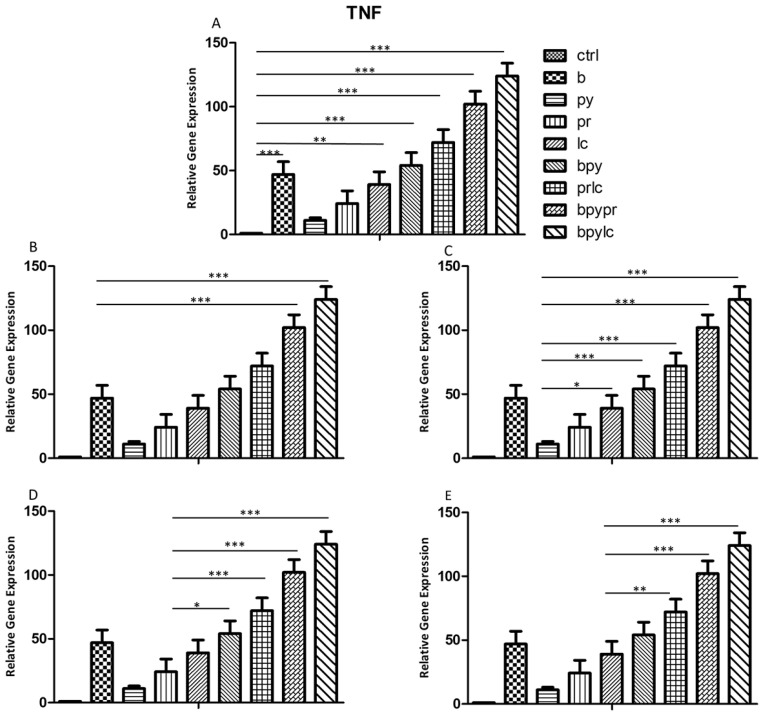
The bar graph shows the statistical analysis, where data are expressed as mean ± standard deviation (SD) from three independent experiments evidencing mRNA levels of *TNF* between ctrl cells (**A**) and each single pesticide b (**B**), py (**C**), pr (**D**), lc (**E**) with the other experimental points. Statistical significance was determined using one-way ANOVA followed by Tukey’s multiple comparison test. Significant differences compared to the control are indicated (* *p* < 0.05, ** *p* < 0.01, *** *p* < 0.001).

**Figure 8 ijms-26-04947-f008:**
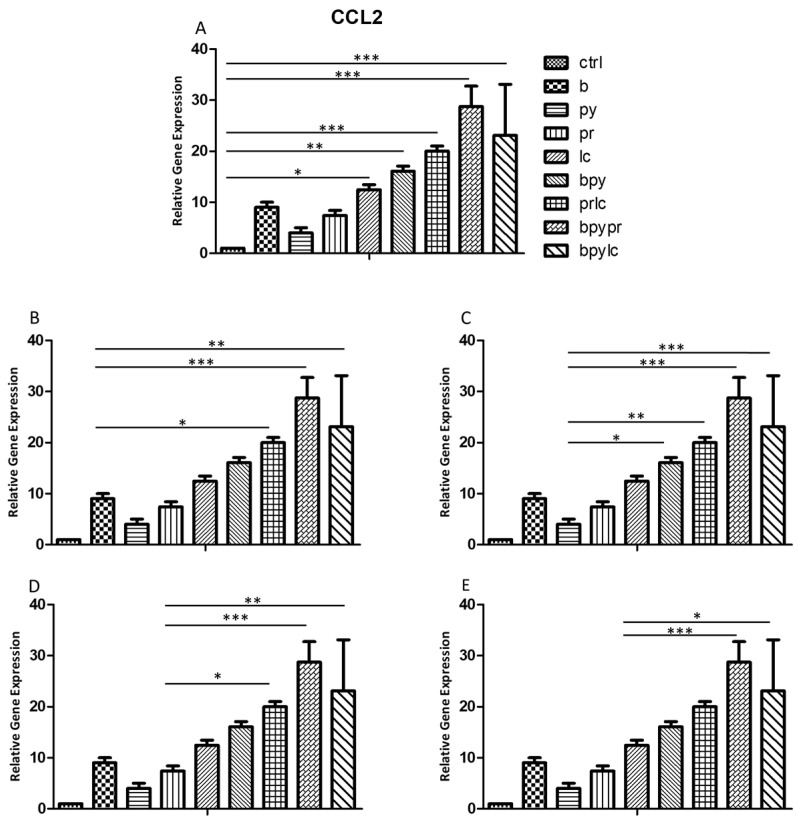
The bar graph shows the statistical analysis, where data are expressed as mean ± standard deviation (SD) from three independent experiments evidencing mRNA levels of *CCL2* between ctrl cells (**A**) and each single pesticide b (**B**), py (**C**), pr (**D**), lc (**E**) with the other experimental points. Statistical significance was determined using one-way ANOVA followed by Tukey’s multiple comparison test. Significant differences compared to the control are indicated (* *p* < 0.05, ** *p* < 0.01, *** *p* < 0.001).

**Figure 9 ijms-26-04947-f009:**
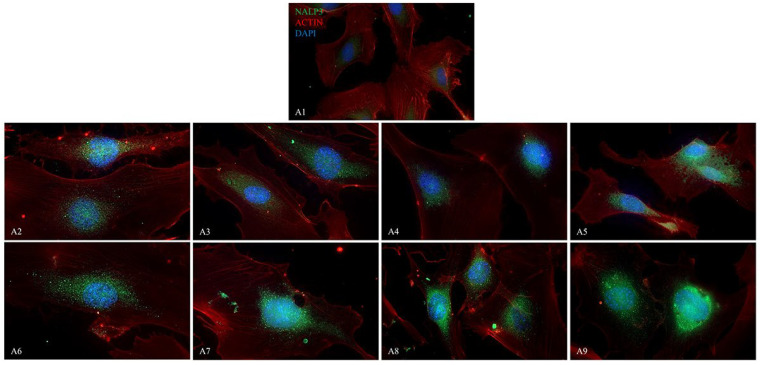
The figure shows the expression of NALP3 analyzed by immunofluorescence microscopy performed in HUVEC cells cultured alone (**A1**), with Boscalid (**A2**), Pyraclostrobin (**A3**), Propamocarb (**A4**), Lamba-cyhalothrin (**A5**), Boscalid + Pyraclostrobin (**A6**), Propamocarb + Lamba-cyhalothrin (**A7**), Boscalid + Pyraclostrobin + Propamocarb (**A8**), and Boscalid + Pyraclostrobin + Lamba-cyhalothrin (**A9**) over a period of 10 days. The experiment was conducted three times, each independently.

**Figure 10 ijms-26-04947-f010:**
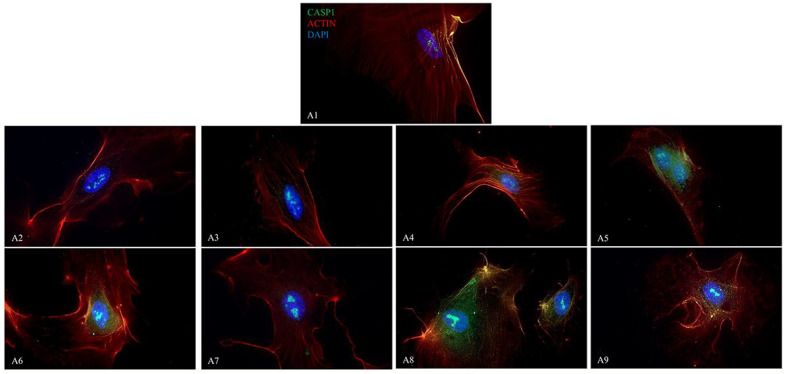
The figures show the expression of CASP1 analyzed by immunofluorescence microscopy performed in HUVEC cells cultured alone (**A1**), with Boscalid (**A2**), Pyraclostrobin (**A3**), Propamocarb (**A4**), Lamba-cyhalothrin (**A5**), Boscalid + Pyraclostrobin (**A6**), Propamocarb + Lamba-cyhalothrin (**A7**), Boscalid + Pyraclostrobin + Propamocarb (**A8**), and Boscalid + Pyraclostrobin + Lamba-cyhalothrin (**A9**) over a period of 10 days. The experiment was conducted three times, each independently.

**Figure 11 ijms-26-04947-f011:**
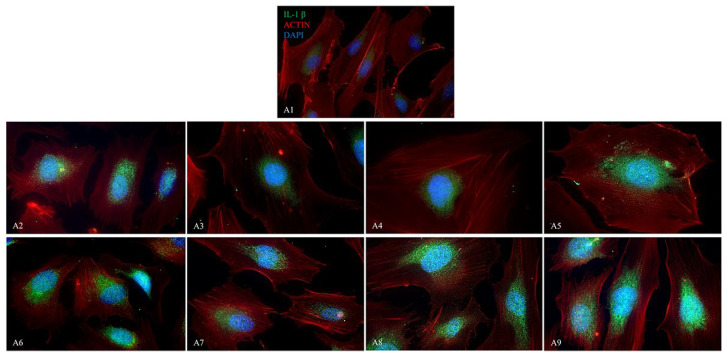
The figure shows the expression of IL-1β analyzed by immunofluorescence microscopy performed in HUVEC cells cultured alone (**A1**), with Boscalid (**A2**), Pyraclostrobin (**A3**), Propamocarb (**A4**), Lamba-cyhalothrin (**A5**), Boscalid + Pyraclostrobin (**A6**), Propamocarb + Lamba-cyhalothrin (**A7**), Boscalid + Pyraclostrobin + Propamocarb (**A8**), and Boscalid + Pyraclostrobin + Lamba-cyhalothrin (**A9**) over a period of 10 days. The experiment was conducted three times, each independently.

**Figure 12 ijms-26-04947-f012:**
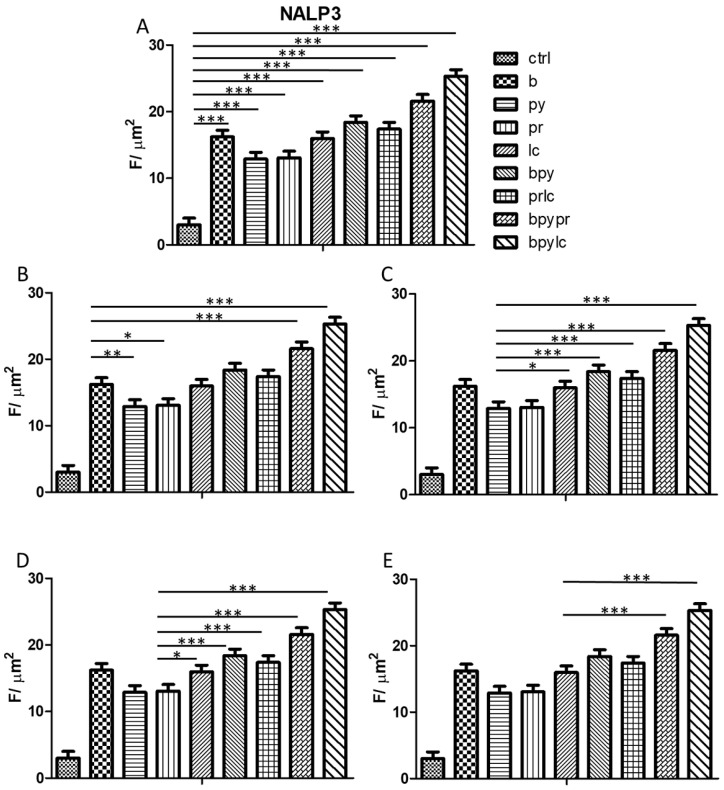
The bar graph shows the statistical analysis, where data are expressed as mean ± standard deviation (SD) from three independent experiments evidencing the NALP3 fluorescence intensity between ctrl cells (**A**) and each single pesticide b (**B**), py (**C**), pr (**D**), lc (**E**) with the other experimental points. Statistical significance was determined using one-way ANOVA followed by Tukey’s multiple comparison test. Significant differences compared to the control are indicated (**p* < 0.05, ** *p* < 0.01, *** *p* < 0.001).

**Figure 13 ijms-26-04947-f013:**
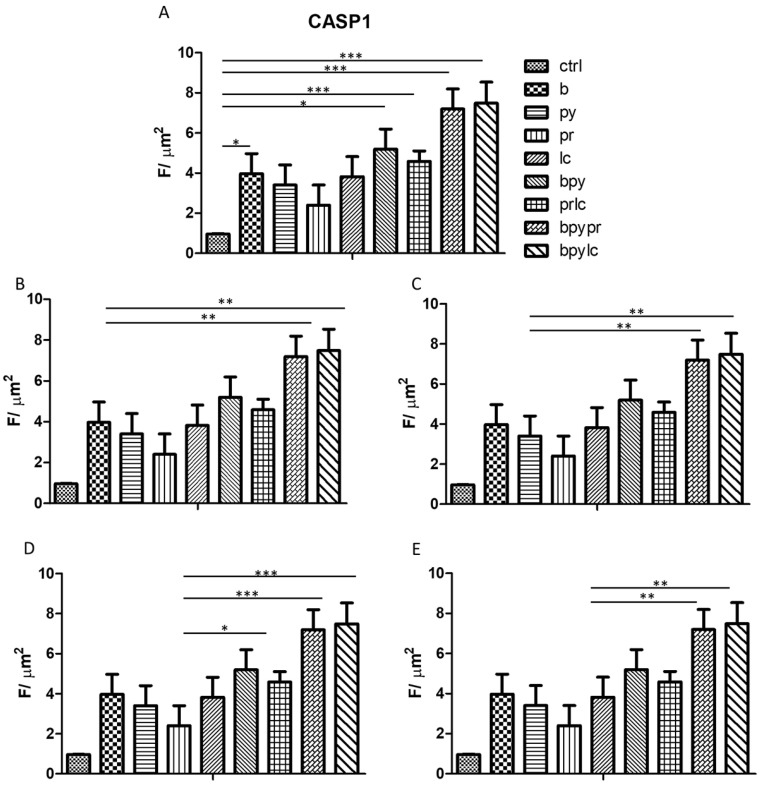
The bar graph shows the statistical analysis, where data are expressed as mean ± standard deviation (SD) from three independent experiments evidencing the CASP1 fluorescence intensity between ctrl cells (**A**) and each single pesticide b (**B**), py (**C**), pr (**D**), lc (**E**) with the other experimental points. Statistical significance was determined using one-way ANOVA followed by Tukey’s multiple comparison test. Significant differences compared to the control are indicated (* *p* < 0.05, ** *p* < 0.01, *** *p* < 0.001).

**Figure 14 ijms-26-04947-f014:**
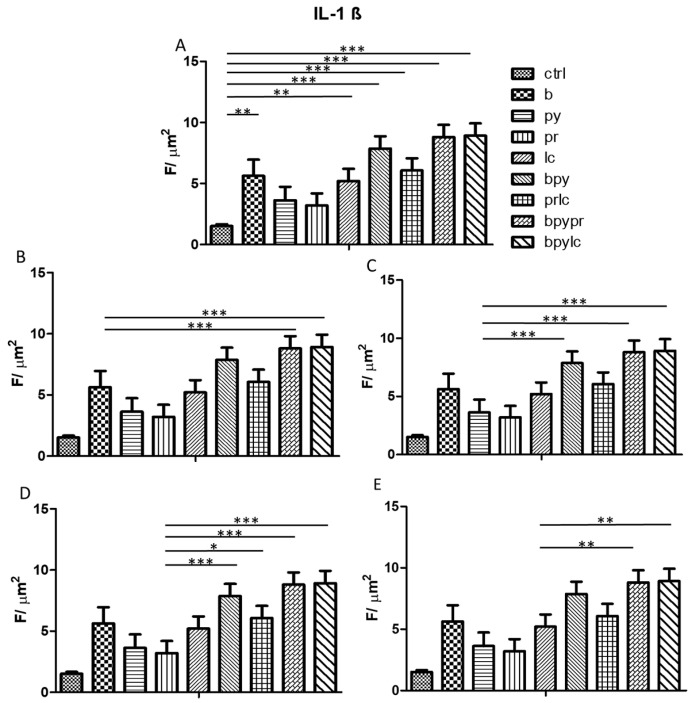
The bar graph shows the statistical analysis, where data are expressed as mean ± standard deviation (SD) from three independent experiments, evidencing IL-1β fluorescence quantification between ctrl cells (**A**) and each single pesticide b (**B**), py (**C**), pr (**D**), lc (**E**) with the other experimental points. Statistical significance was determined using one-way ANOVA followed by Tukey’s multiple comparison test. Significant differences compared to the control are indicated (* *p* < 0.05, ** *p* < 0.01, *** *p* < 0.001).

**Figure 15 ijms-26-04947-f015:**
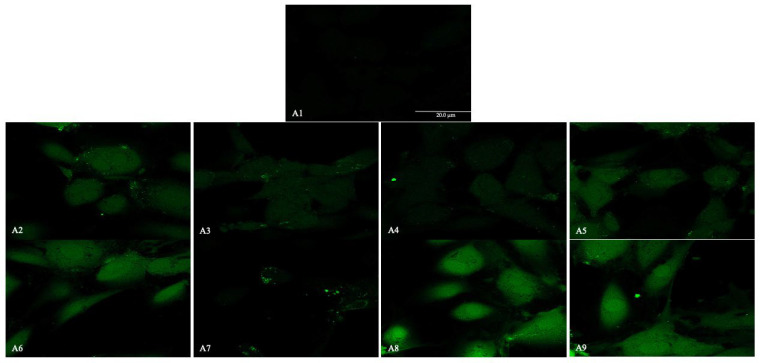
The figure shows ROS measurements from live cells filled with DCFH-DA and captured using confocal microscopy. In particular, the pictures show HUVECs cultured in different conditions: alone (**A1**), with Boscalid (**A2**), with Pyraclostrobin (**A3**), with Propamocarb (**A4**), with Lambda-cyhalothrin (**A5**), with Boscalid + Pyraclostrobin (**A6**), with Propamocarb + Lambda-cyhalothrin (**A7**), with Boscalid + Pyraclostrobin + Propamocarb (**A8**), and with Boscalid + Pyraclostrobin + Lambda-cyhalothrin (**A9**). The scale bar measures 20 micrometers. The experiment was conducted in triplicate.

**Figure 16 ijms-26-04947-f016:**
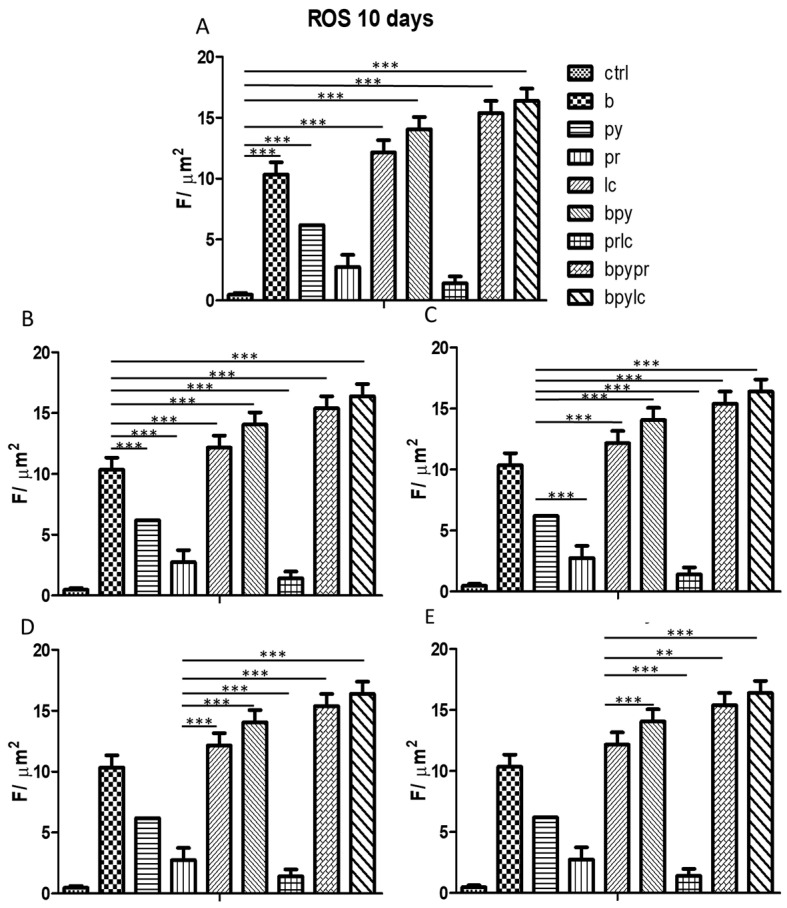
The bar graph shows the statistical analysis, where data are expressed as mean ± standard deviation (SD) from three independent experiments, evidencing the production of ROS levels between ctrl cells (**A**) and each single pesticide b (**B**), py (**C**), pr (**D**), lc (**E**) with the other experimental points. Statistical significance was determined using one-way ANOVA followed by Tukey’s multiple comparison test. Significant differences compared to the control are indicated (** *p* < 0.01, *** *p* < 0.001).

**Figure 17 ijms-26-04947-f017:**
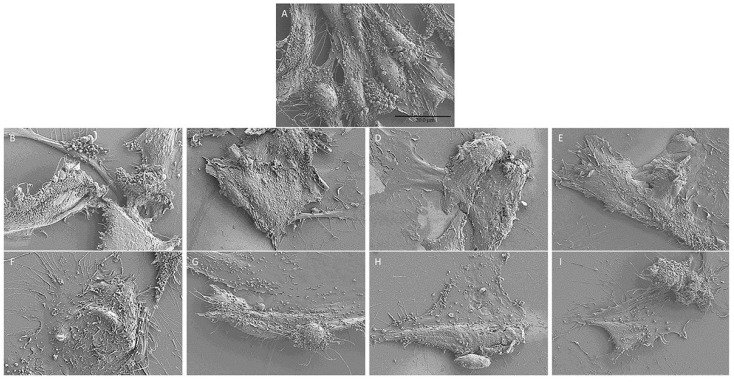
SEM pictures show morphological analysis in HUVEC control (**A**) compared with HUVECs exposed to b (**B**), py (**C**), pr (**D**), lc (**E**), b + py (**F**), pr + lc (**G**), b + py + pr (**H**), b + py + lc (**I**) after 10 days of treatment. The experiment was conducted three times.

**Table 1 ijms-26-04947-t001:** Pesticides type, mode of action, and hNDI values.

Pesticide	Tipology	Mode of Action	hNDI (μM)
Boscalid	Fungicide	Succinate dehydrogenase (SDH) inhibition in the mitochondrial electron transport chain	11.06
Pyraclostrobin	Fungicide	Single-site mode of action, inhibition of mitochondrial respiration	1.51
Propamocarb	Fungicide	Antisporulating activity, interferes in the synthesis of fatty acids and phospholipids, damaging the formation of cell membranes.	15.64
Lambda-cyhalothrin	Insetticide	Action on the central and peripheral nervous system, at the level of axonal conduction with alteration of the permeability of the neuron membrane	0.19

**Table 2 ijms-26-04947-t002:** Pesticide combination.

Combination	Code
Boscalid Pyraclostrobin	b + py
Boscalid+ Pyraclostrobin + Propamocarb	b + py + pr
Propamocarb + Lambda-cyhalothrin	pr + lc
Boscalid+ Pyraclostrobin + Lambda cyhalothrin	b + py + lc

**Table 3 ijms-26-04947-t003:** Primer sequences used for real-time PCR reactions.

Gene	Forward Primer (5′-3′)	Revers Primer (5′-3′)
*NALP3*	5′-GAATGCCTTGG-GAGACTCAG-3′	5′-AGATTCTGATT-AGTGCTGAGTACC-3′
*CASP1*	5′-TGCCTGTTCCTGTGATGTG-3′	5′-GTAGAAACATCTTGTCAAAGTCACT-3′
*IL-1β*	5′-CGTCCTAAAGA-CTCCATGATCTG-3′	5′-ACCAATCTTGT-AGGACTGACC-3′
*IL-6*	5′-GCAGATGAGTACAAAAGTCCTGA3′	5′-TTCTGTGCCTGCAGCTTC-3′
*CCL2*	5′-AGCAGCCACCTTCATTCC-3′	5′-GCCTCTGCACTGAGATCTTC3′
*TNF*	5′-TGCACTTTGGAGTGATCGG-3′	5′-TCAGCTTGAGGGTTTGCTAC-3′
*GAPDH*	5′-ACATCGCTCAGACACCATG-3′	5′-TGTAGTTGAGGTCAATGAAGGG-3′

## Data Availability

Data are available to the corresponding author upon request.

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
