# Peer review of "Activation of the NALP3-CASP1-IL-1 β Inflammatory Pathway by Pesticide Exposure in Human Umbilical Vein Endothelial Cells"

_ijms, 2025, doi:10.3390/ijms26104947_

Round 1
Reviewer 1 Report
Comments and Suggestions for Authors
This study describes effects of several widely used pesticides on survival and proinflammatory activity of endothelial cells. Using a tetrazolium salt assay to quantify the cell metabolic activity, the authors demonstrated toxicity after long-term (10 days) exposure of HUVECs to physiologically relevant doses of pesticides. They further used fluorescent microscopy to demonstrate pesticide-induced elevated levels of NALP3, caspase-1, and pro-IL-1b mRNA and proteins induced by pesticides, as well as formation of NALP3-containing specks suggesting the assembly of active inflammasomes. Finally, they demonstrated development of oxidative stress induced by pesticides, and described morphological changes in cells including formation of blebs. To summarize, the authors demonstrated classical signs of endothelial activation and damage includung oxidative stress, reduction of metabolic activity, and proinflammatory changes, in this case assembly of inflammasomes.
Simple experiments could improve the ms. First, to confirm activation of inflammasomes, IL-1b (mature) ELISA can be used. RT-PCR can be used to demonstrate induction of other pro-inflammatory mediators and leukocyte adhesion molecules such as IL-8, MCP-1, E-selectin and VCAM-1. Because endothelial damage increases its pro-coagulant properties, I suggest measuring tissue factor activity in a clotting assay or by mRNA expression. A drawback of the current version of the ms is that single-dose measurements produce quite limited information. At least for simple and inexpensive tests such as MTP, I suggest to produce concentration curves. It is also very important to describe selection of concentrations in more details and compare selected concentrations with data of other authors and, maybe, with the results of in vivo analytical studies. How ADME was taken into account to calculate bioavailability should be better explained. It seems to be taken as 100%, which is not realistic and may result in testing in vitro concentrations that are never reached in vivo. Please check it.
There are multiple mistakes in figure numbering. The language has to be corrected. DMSO concentrations were not described.
Comments on the Quality of English LanguageMultiple small mistakes
Author Response
Comment 1: This study describes effects of several widely used pesticides on survival and proinflammatory activity of endothelial cells. Using a tetrazolium salt assay to quantify the cell metabolic activity, the authors demonstrated toxicity after long-term (10 days) exposure of HUVECs to physiologically relevant doses of pesticides. They further used fluorescent microscopy to demonstrate pesticide-induced elevated levels of NALP3, caspase-1, and pro-IL-1b mRNA and proteins induced by pesticides, as well as formation of NALP3-containing specks suggesting the assembly of active inflammasomes. Finally, they demonstrated development of oxidative stress induced by pesticides, and described morphological changes in cells including formation of blebs. To summarize, the authors demonstrated classical signs of endothelial activation and damage includung oxidative stress, reduction of metabolic activity, and proinflammatory changes, in this case assembly of inflammasomes.
Simple experiments could improve the ms. First, to confirm activation of inflammasomes, IL-1b (mature) ELISA can be used. RT-PCR can be used to demonstrate induction of other pro-inflammatory mediators and leukocyte adhesion molecules such as IL-8, MCP-1, E-selectin and VCAM-1. Because endothelial damage increases its pro-coagulant properties, I suggest measuring tissue factor activity in a clotting assay or by mRNA expression. A drawback of the current version of the ms is that single-dose measurements produce quite limited information. At least for simple and inexpensive tests such as MTP, I suggest to produce concentration curves. It is also very important to describe selection of concentrations in more details and compare selected concentrations with data of other authors and, maybe, with the results of in vivo analytical studies. How ADME was taken into account to calculate bioavailability should be better explained. It seems to be taken as 100%, which is not realistic and may result in testing in vitro concentrations that are never reached in vivo. Please check it.
There are multiple mistakes in figure numbering. The language has to be corrected. DMSO concentrations were not described.
Answers: We thank the Reviewer for having spent time to revise our study.
We thank for the suggestion regarding the use of ELISA to confirm activation of inflammasomes, IL-1b; we will proceed with applying this assay in a future study to further investigate the activation of inflammation by pesticide exposure.
However, we improved our manuscript by adding results regarding the expression of the mRNA, obtained via RT-PCR, of the markers CCL-2 (MCP-1) and N-Cad, chosen as leukocyte adhesion molecules, and IL6 and TNF, chosen as other pro-inflammatory molecules.
We aimed at analyzing those markers because, according to the literature, CCL2 (a chemokine) and N-Cadherin (a cadherin) play distinct but related roles in leukocyte adhesion and migration. CCL2, also known as MCP-1, acts as a chemoattractant, attracting leukocytes to sites of inflammation. N-Cadherin, a cell adhesion molecule, is involved in cell-cell adhesion, including between leukocytes and endothelial cells. While N-Cadherin helps form junctions, CCL2 helps to recruit leukocytes and potentially disrupt these junctions, facilitating leukocyte migration. (Monocyte Chemoattractant Protein-1 (MCP-1): An Overview; Extravasation of leukocytes in comparison to tumor cells)
Moreover, a recent study demonstrated that in the absence of VE-cadherin, N-cadherin can replace the function of VE-cadherin in the adherens junctions. VE-cadherin is important for the inflammation phenome because it moves laterally in the membrane so that it is out of the location of the leukocyte migration (Differential Localization of VE- and N-Cadherins in Human Endothelial Cells: VE-Cadherin Competes with N-Cadherin for Junctional Localization)
In addition, IL-6 and tumour necrosis factor (TNF) are key pro-inflammatory cytokines involved in the inflammatory response, playing a significant role in various diseases. IL-6 is a critical mediator in acute inflammation while TNF-α is considered a primary trigger of the inflammatory cascade (Pro-inflammatory cytokines/chemokines, TNF-α, IL-6 and MCP-1, as biomarkers for the nephro and pneumoprotective effect of silibinin after hepatic ischemia/reperfusion: Confirmation by immunohistochemistry and qRT-PCR; The proinflammatory cytokines TNF-α and IL-6 in lumpfish (Cyclopterus lumpus L.) -identification, molecular characterization, phylogeny and gene expression analyses)
Moreover, the literature suggests a link between the markers IL1 β, IL-6 and TNF; in fact, according to the study “Inflammasome activation at the crux of severe COVID-19” the activation of IL-1β, released by inflammasome signalling, activates monocytes, which can lead to the secretion of both IL-6 and TNF.
The switch-on of these cytokines, then, can cause inflammation through various mechanisms, including recruitment of neutrophils; in fact, IL-1β and IL-6 can downregulate adherens junctions in endothelial cells, which increases their permeability and could contribute to coagulation in the vasculature. In addition, Neumann et al demonstrated that IL-6 induces the membrane protein tissue factor (TF) on the surfaces of monocytes (Effect of human recombinant interleukin-6 and interleukin-8 on monocyte procoagulant activity). TF promotes activation of the coagulation system by initiating the extrinsic coagulation pathway. It activates factors VIIa and Xa, following activating prothrombin to produce thrombin that cleaves fibrinogen to fibrin to generate fibrin clot formation. In an attempt to administer recombinant IL-6 to humans, IL-6 was shown to stimulate coagulation through an increase in thrombin-antithrombin III complexes and the prothrombin activation fragment F1 + 2. Moreover, as shown in a trial in healthy humans, the activation of coagulation through the administration of recombinant factor VIIa elicited an increase in plasma IL-6 (Activation of coagulation by administration of recombinant factor VIIa elicits interleukin 6 (IL-6) and IL-8 release in healthy human subjects).
Thus, the confirmed activation of the pro-inflammatory markers IL-1β, IL6 and TNF suggest the correlated activation of the procoagulation process.
In a future study, we will aim to measure the tissue factor activity itself by mRNA expression; we thank the Reviewer for the suggestion.
Regarding the request of MTP, we want to underline that we performed the MTS assay that measures cell viability by detecting the metabolic activity of cell; we kindly think that MTP is not suitable with the aim of our study since the assay focuses on phospholipid transfer, that does not match the focus of our study.
However, we improved our manuscript by showing concentration curves derived from the cell count obtained with the use of the trypan blue solution.
We understand the importance of comparing our data, from the calculation of the pesticides concentrations to the obtained results, with data of other authors, but the truth is that our study is the first to investigate these pesticides in a single dose and in the chosen combination at hNDI concentration, so the comparison is difficult. This represents a peculiarity of our manuscript being the first to fill the gap described in our study regarding the missing investigation of the effects of the most common European pesticides.
We also revised the text regarding ADME and DMSO considerations based of your comment.
Additionally, the request regarding the quite limited information linked to the single-dose measurements is not so clear since we presented the same results reported for the combined pesticides.
We thanks for the kind comment regarding the figures and the language; we managed to revise these points. However, we want to underline that the manuscript has been revised by Dr. Antonio Nanci, who is a native language and full Professor from the Universitè dè Montrèal.
Reviewer 2 Report
Comments and Suggestions for Authors
Data based on only HUVECs can be accepted but their scientific value is very limited. If no mice model used, then at least some conclusions would be better to verify using some other cell types preferably both endothelial and non-endothelial.
Lines 112-113: the sentence (highlighted in blue) seems to be not complete.
I would suggest some more description of NALP3/CASP1/IL-1β pathway in the Introduction (not in Discussion). Do HUVECs show signs of pyroptosis shown in the literature?
Figures 2-4: staining type should be indicated directly on the figure (under or above the panels), not in Figure legends. I would also recommend to make distance between the panels much smaller. The same suggestion is for the other figures as well.
Figures 5-7: In the figure legend there is no mention what does (A) - (E) mean. The reader have to guess...
Figures 8-10, 12: There is not even (A), (B),... indications on the figure panels! Not to say a word about figure legends, which is already "traditionally" lacking this description.
Why did the authors performed RT-qPCR analysis (chapter 2.3)? They already showed IFL data. What was the point for RT-qPCR? To validate IFL results (as the authors mention - line 234)? Usually people want validate RNA data with IFL staining but not vice versa. In this cell model RT-qPCR should be used not for validation of IFL staining data, but to measure more molecular types thaty can be invoilved in the inflammatory responses. As a first step, I would recommend to run some kind of RNA profiling using commercially available kits, for example, from Qiagen. Then the selected RNA species can be verified by RT-qPCR.
There is no justification for ROS measurements (chapter 2.4). For example, why should we expect alterations of ROS in pesticides-treated cells.
No quantification (and statistics) of ROS probe IFL; only representative images are shown.
Some parts in Discussion simply repeat the same thing, already mentioned in Introduction (example - lines 280 - 295).
MTS method is good but it doesn't provide clear understanding what's going on in the cell culture. If cells do not proliferate well, they are more sparse, compared to control, and the colored MTS signal will be apparently less. The same conclusion can be made if cells die more actively or if they have lower metabolic actgivity due to some depression. At least cell counts shound be shown in parallel to MTS.
If the authors show accumulation of NALP3, Casp1 and IL-1beta, it does not directly tells us that HUVECs are dying due to pyroptosis. The authors did not measure any forms of cell death, such as pyroptosis or apoptosis. I guess it should be done.
Then, there is an obvious question: if the authors have used hNDI values of pesticides (derived from daily consumption by humans) for the treatment of HUVECs, and they show clear effects in HUVEC cultures, I'm wondering why the people in French or Italian population do not suffer much from the intake of pesticides that comes to their organisms with the vegetables? Why are HUVECs so sensitive? Should the authors test some other (endothelial) cell types, such HAECs, which are available commercially, for example in PromoCell? I should mention here that HUVECs are not good as model system, because, unlike their conditions in umbilical vein in vivo, in cell culture in vitro they actively proliferate, weakly attach to plastic and basically show high sensitivity to various factors, compared to many other endopthelial and non-endothelial cell types.
In Discussion the authors provide some overview of extracellular vesicles in endothelial physiology. But they don't show any relevant data on that point. And this piece appears a little strange because there is no clear ideas what would be the vesicles released fropm the pesticides-treated HUVECs? Would they contain IL-1beta and IL-18 or something else? Should this point be studied more deeply by the authors?
Author Response
Comment 1: Data based on only HUVECs can be accepted but their scientific value is very limited. If no mice model used, then at least some conclusions would be better to verify using some other cell types preferably both endothelial and non-endothelial.
Lines 112-113: the sentence (highlighted in blue) seems to be not complete.
I would suggest some more description of NALP3/CASP1/IL-1β pathway in the Introduction (not in Discussion). Do HUVECs show signs of pyroptosis shown in the literature?
Answer 1: Thanks to the Reviewer for the kind suggestion; we improved the part of the NALP3/CASP1/IL-1β pathway that is described as pyroptosis itself; thus, we can confirm, as added in the manuscript, that the activation of those markers suggests that the pesticide could activate the pyroptosis. We also thank for suggesting using another cell line; it was already a plan for our future study to understand the role of the pesticides on another endotelial line to confirm the obtained data and compare our results.
Comment 2: Figures 2-4: staining type should be indicated directly on the figure (under or above the panels), not in Figure legends. I would also recommend to make distance between the panels much smaller. The same suggestion is for the other figures as well.
Answer 2: Thanks to the Reviewer for the suggestion; we have modified the figures as suggested.
Comment 3: Figures 5-7: In the figure legend there is no mention what does (A) - (E) mean. The reader have to guess...
Answer 3: Thanks to the Reviewer for the suggestion
Comment 4: Figures 8-10, 12: There is not even (A), (B),... indications on the figure panels! Not to say a word about figure legends, which is already "traditionally" lacking this description.
Answer 4: Thanks to the Reviewer for the suggestion
Comment 5: Why did the authors performed RT-qPCR analysis (chapter 2.3)? They already showed IFL data. What was the point for RT-qPCR? To validate IFL results (as the authors mention - line 234)? Usually people want validate RNA data with IFL staining but not vice versa. In this cell model RT-qPCR should be used not for validation of IFL staining data, but to measure more molecular types thaty can be invoilved in the inflammatory responses. As a first step, I would recommend to run some kind of RNA profiling using commercially available kits, for example, from Qiagen. Then the selected RNA species can be verified by RT-qPCR.
Answer 5: We thank to the Reviewer for the kind suggestion; we agree with the IFL confirming the RT-PCR findings. Thus, we revised the manuscript and the disposition of the results to resolve this misunderstanding.
Moreover, following your suggestion, we used RT-PCR to validate other molecules involved in inflammatory responses, as IL-6 and TNF; of course, we amplified the manuscript by describing the roles of these molecules and justifying their choice, showing the relating results obtained.
Comment 6: There is no justification for ROS measurements (chapter 2.4). For example, why should we expect alterations of ROS in pesticide-treated cells.
Answer 6: Thanks to the Reviewer for the kind suggestion; we provided amplification to this part.
Comment 7: No quantification (and statistics) of ROS probe IFL; only representative images are shown.
Answer 7: The figure 17 shows the quantification of ROS with the relative statistical analysis; the text also report how we measured the ROS expression fluorescence and the statistical test used.
Comment 8: Some parts in Discussion simply repeat the same thing, already mentioned in Introduction (example - lines 280 - 295).
Answer 8: Thanks to the Reviewer for the kind suggestion; we provided to modify this part.
Comment 9: MTS method is good but it doesn't provide clear understanding what's going on in the cell culture. If cells do not proliferate well, they are more sparse, compared to control, and the colored MTS signal will be apparently less. The same conclusion can be made if cells die more actively or if they have lower metabolic activity due to some depression. At least cell counts should be shown in parallel to MTS.
Answer 9: Thanks to the Referee for this point, we improved our manuscript by adding results derived from cell counts obtained by trypan blue assay to provide furthermore information about pesticide exposure on cell proliferation.
Comment 10: If the authors show accumulation of NALP3, Casp1 and IL-1beta, it does not directly tells us that HUVECs are dying due to pyroptosis. The authors did not measure any forms of cell death, such as pyroptosis or apoptosis. I guess it should be done.
Answer 10: Thanks to the Review for this comment; as we reported in the revised manuscript, the NALP3, Casp1 and IL-1beta are part of the typical pathway identified as pyroptosis. Thus, the findings obtained from our study suggest the activation of this particular cell death induced by pesticide exposure. We are currently setting up new investigations on treated pesticide cells to confirm this hypothesis and to find out if there are other cell death pathways activated by those pollutants
Comment 11: Then, there is an obvious question: if the authors have used hNDI values of pesticides (derived from daily consumption by humans) for the treatment of HUVECs, and they show clear effects in HUVEC cultures, I'm wondering why the people in French or Italian population do not suffer much from the intake of pesticides that comes to their organisms with the vegetables? Why are HUVECs so sensitive? Should the authors test some other (endothelial) cell types, such HAECs, which are available commercially, for example in PromoCell? I should mention here that HUVECs are not good as model system, because, unlike their conditions in umbilical vein in vivo, in cell culture in vitro they actively proliferate, weakly attach to plastic and basically show high sensitivity to various factors, compared to many other endopthelial and non-endothelial cell types.
Answer 11: Thanks to the Referee for underlining this point.
HUVECS cells have been chosen as an in vitro model to represent the endothelial tissue for all the reasons already provided in the manuscript; of course, like the others in vitro models, the 2d model can’t represent the entire organism and all the complicated mechanisms associated. However, many articles in the literature clearly describe the various negative effects on the French and Italian populations. (The social costs of pesticide use in France; Health status of Italian children living close to cultivations sprayed with pesticides; Pesticides: what are the risks to our health and to the environment?) Of course, as reported in our manuscript, the use of pesticides is an issue that involves the entire world; our study, however, is focused on the European population.
Our investigation on pesticides effects are not limited to this model; we are currently setting up other experiments to focus on other cell lines to compare the results already obtained with the HUVECs model. We thanks the Referee for the cell line suggested; however, we did not face the problems mentioned by the Referee studying the HUVEC. They attached to the plastic without any kind of problem and regarding the sensitivity, we compared the treated cells with the control in order to normalize the results obtained.
Comment 12: In Discussion the authors provide some overview of extracellular vesicles in endothelial physiology. But they don't show any relevant data on that point. And this piece appears a little strange because there is no clear ideas what would be the vesicles released from the pesticides-treated HUVECs? Would they contain IL-1beta and IL-18 or something else? Should this point be studied more deeply by the authors?
Answer 12: We thank the Reviewer for this point; we decided to report our findings about the alterations of the extracellular vesicles, observed in SEM pictures, as an important starting point that brought us to deeply investigate their profile in a future study. We are organizing to identify the proteic profile inside those vesicles to better understand their role in the activation of the inflammatory pathway, such as other pathways. To clarify this point, we revised the manuscript so people can understand that in a future study we will also show this other informations.
However, regarding the improvement of the English, we want to underline that the manuscript has been revised by Dr. Antonio Nanci, who is a native language and full Professor from the Universitè dè Montrèal.
Round 2
Reviewer 1 Report
Comments and Suggestions for Authors
- Trypan Blue is not a test for cell count, it is a test for proportion of dead cells. Correct figure 2.
- in figures from 3 to the last one I see identical panels A-E, except the statistical significance is different. Please check
- figure legends are mostly unclear. Please check. And don't mention 3 (!) times in each legend that the data are presented as mean +/- SD
- N-cadherin is not a typical inflammation-induced leukocyte adhesion molecule. i would remove these data from the ms.
Author Response
- Trypan Blue is not a test for cell count; it is a test for proportion of dead cells. Correct figure 2.
Thanks to the Referee for the kind suggestion, we provided at modifying the figure 2
- In figures from 3 to the last one I see identical panels A-E, except the statistical significance is different. Please check
Thanks to the Refere for this point.
Each figure represents the same panel from A to E because in the panel A is the ctrl to be compared with the other experimental points, while in panel B is the pesticide Boscalid, in panel C is Pyraclostrobim, in figure D is Propamocarb and in panel E is the Lambda-cyhalothrin. We made this decision to make the statistical analysis as clear as possible. The panels seem to be similar to each other because the trend of the pesticides was very coherent, but in the specific, they are different.
- Figure legends are mostly unclear. Please check. And don't mention 3 (!) times in each legend that the data are presented as mean +/- SD
Thanks to the Referee for the kind suggestion, we provided at modify the figure legend.
- N-cadherin is not a typical inflammation-induced leukocyte adhesion molecule. i would remove these data from the ms.
Thanks to the Referee for the kind suggestion, we provided at removing the data
Reviewer 2 Report
Comments and Suggestions for Authors
No additional comments. The authors addressed my concerns.
Author Response
No additional comments. The authors addressed my concerns.
Thanks to the Referee for spending time to review our manuscript
Round 3
Reviewer 1 Report
Comments and Suggestions for Authors
The manuscript is still full with typos and language mistakes. please use AI to correct.
Comments on the Quality of English Languagelanguage must be corrected, i suggest AI